# How the Digital Economy Enables Regional Sustainable Development Using Big Data Analytics

**Ruohan Wang [1], Qingjin Wang [2,\*], Renbo Shi [3], Kaiyun Zhang [2] and Xueling Wang [2]**

[1] School of Business and Economics, Namseoul University, Cheonan 31020, Republic of Korea; wrh51777@163.com
[2] Business School, Qingdao University, Qingdao 266071, China; 17854212332@163.com (K.Z.); icherlynw@icloud.com (X.W.)
[3] School of Economic and Management, Beihang University, Beijing 100083, China; renboshi0705@163.com
\* Correspondence: 2020021206@qdu.edu.cn; Tel.: +86-15653231261

**Abstract:** The development of the cultural industry cannot be isolated from the efficient integration with the digital economy and digital technology at the current stage of the technological and industrial revolution. This paper constructs an indicator system to measure the sustainable development of the cultural industry and tests the relationship between the digital economy and the sustainable development of the cultural industry using an OLS model based on China's provincial panel data from 2011 to 2021. The findings of this study suggest that the digital economy can significantly aid in the long-term growth of cultural companies. The process of promoting sustainable development of the cultural industry through the digital economy has also advanced thanks to the government's strong support. This report also suggests governmental recommendations based on these findings for the sustainable development of China's cultural industry in the age of the digital economy. This paper theoretically elucidates the mechanism of the role of the digital economy on the sustainable development of the cultural industry, constructs a system of indicators to measure the sustainable development of the cultural industry, and tests the impact of the digital economy on the sustainable development of the cultural industry.

**Keywords:** digital economy; sustainable development; cultural industry

## 1. Introduction

The introduction of numerous digital technologies, including artificial intelligence and digital twins, has made the digital economy a new driving force for promoting China's economic growth. A favorable technological, informational, and financial environment for businesses' production and operation activities can be produced by the distinctive qualities of disruptive technological innovation, information transparency, and low-cost sharing. The impact of the digital economy on industrial structure, production efficiency, employment, innovation, economic growth, sustainable development, and other elements has been the subject of much qualitative research [1]. It argues that the digital economy is capable of disrupting ideological patterns at different scales from micro to macro and promoting industrial integration and economic efficiency. The information technology innovation in the digital economy has created more effective channels for information sharing, encouraging industrial innovation and economic structural transformation [2].

Diverse consumption patterns have emerged recently as a result of the need for a prolonged recovery of the global economy, and industrial development is currently seeking transformation and upgrading. The digital culture industry's growth potential is continually emphasized. Digital cultural industries rely more on technical transformation, improvement, and application integration than traditional cultural businesses. The limitations of "text" creation and "creative" production in terms of time and space are further shattered, and the economic formats that are gathered around "data" are more obvious [3].

In addition to fostering cross-regional, cross-hierarchical, and cross-sectoral collaboration in the digital culture industry, the creation of a unified national market can also increase the long tail effect and the beehive effect and enable dual circulation. The entire process of symbol creation by symbol creators and symbol consumption by symbol consumers is the focus of the digital culture industry, which emphasizes intelligent workflow and promotes the use of AI and automation technology in daily operations to improve responsiveness and execution, as well as the integration of highly sensitive digital operations in a single market [4]. High-precision digital technology is needed to build a digital platform that can allow real-time perception of change, real-time analysis of change, real-time formulation of best decisions, and automatic implementation of decisions. In particular, it can break down regional barriers and market segmentation from the outside, enabling regional collaborative development. It can also remove barriers to the movement of commodities within the industry.

The national cultural system is carried by the cultural industry, which is a significant part of the national economic industrial system. Promoting the cultural sector's sustainable growth serves two purposes: it fosters high-quality economic growth and increases cultural pride and self-improvement. The "China Cultural Industry Investment and Financing Report (2021)" demonstrates that although the general investment and financing position of China's culture industry is improving, there are still significant problems, including uneven socioeconomic gains and shoddy industrial investment. In the process of transforming the digital economy from consumption to production, the penetration of new technologies has created new forms and models of cultural consumption and has become a new driving force for the sustainable development of the cultural industry. Therefore, it is of significant theoretical value to describe the boosting mechanism of the digital economy in the sustainable development of the cultural sector given the general trend of the rapid development of digital technologies such as blockchain, big data, and 5G networks. In addition, existing studies have paid extensive attention to the impact of cultural innovation [5], talent cultivation [6], branding [7], industrial integration [8], and other factors on the sustainable development of the cultural industry. While existing research provides important insights into how to promote the sustainable development of cultural industries, few scholars have explored the impact on the sustainable development of cultural industries from the perspective of the digital economy and big data analytics. The next section of this study will analyze how the digital economy supports the sustainable development of the cultural sector and look for workable implementation strategies to achieve that sector's sustainable development.

## 2. Literature Review

The cultural industry is a global priority for all nations [9], but in contrast to developed nations, China still lags far behind in the five areas of capital investment, production efficiency, industrial scale, policies and regulations, and technological innovation [10]. China's current policy system for the cultural industry is still not perfect enough, and the targeting and operability are not strong [11]. In contrast, the United States, South Korea, and other nations regulate and promote the development of the cultural industry through legislation and other measures and have formed a complete legal system for the protection of intellectual property rights, movie grading and rating systems, etc. [12]. Furthermore, China's cultural industry has developed more quickly than developed nations, which prevents it from achieving the inclusive development of cultural plurality [13]. Additionally, given how digital technology has affected China's traditional cultural industry, China's digital cultural industry has a greater need for institutional innovation in order to support the long-term production of cultural creativity in China's cultural industry [14].

Because China's cultural sector is undergoing a digital transformation, it serves as an excellent case study for the noteworthy characteristics of the new digital cultural sector, including significant clustering, connectivity [15], virtualization [16], challenges related to humanism and ethics [17], cultural experience differences [18], and regional and spatial organization. The growth of China's cultural industry offers numerous examples and rich

practices for the transformation and study of the global cultural industry [19]. Therefore, we will take the sustainable development of China's cultural industry as the research object and explore the influence mechanism of the digital economy on the sustainable development of China's cultural industry.

## 3. Research Hypothesis

### 3.1. Digital Economy and Sustainable Development in the Cultural Industry

The transformation and use of digital technology have become the most fundamental cause of economic and social transformation [20]. The high permeability and high participation characteristics of digital technology can generate technology diffusion effects in multiple enterprises. All links have been penetrated, the type and proportion of factor input into the production process have gradually changed, the breadth and depth of information exchange between regions has increased, the cross-regional integration of capital and factors has been promoted, the traditional market constraints have been broken, and resource mismatch and market distortion have been reduced [21]. The digital transformation of industry can connect the originally scattered equipment, enterprises, markets, etc., not only to realize the linked development in R&D, production, supply chains, and market within the enterprise, but also to enhance the innovation performance of enterprises, change their innovation methods and types of innovation, and expand the overall performance of industry by strengthening the connection and interoperability between different enterprises and between enterprises and markets [22]. The empowering effect of digitization on economic growth is becoming increasingly evident, and this has brought about changes in various aspects such as production factors, industrial patterns, and development models.

Cultural industries, as a special cultural phenomenon and an economic tool, influence people's grasp of the essence of culture, and different countries have different understandings of cultural industries from different perspectives. However, China's cultural sector is still not widely accepted from the standpoint of the industrial division of labor, and the high-innovation and high-value aspects of the industrial chain still mainly lie in developed countries. Moving from large-scale to high value is the inevitable path of sustainable development in cultural digitalization. The digitization of the cultural industry is a unique cultural phenomenon of a digital society that emerged with the development of digital technology and the internet, and brand-new production methods, lifestyles, and ways of thinking have been created for human beings via modern information technology [23]. The digital economy has the potential to support the sustainable development of the cultural sector through the provision of digital technologies and inclusive finance.

First off, new technologies like "5G + 8K", artificial intelligence, and virtual reality are developing quickly thanks to the active development of the next generation of network information technology. Digital technology has demonstrated strong vigor and vitality, emerging as a significant driving force for resolving the structural conflict between talent skills and enterprise needs in the labor market and enabling high-quality economic development. Digital technology has a transformative and upgrading impact on the cultural industry that goes beyond simple technology superposition and upgrading to include a deep integration across a number of linkages including design, operation, marketing, and consumption. The explosive growth of cultural digital products and the lifting of spatial and temporal constraints on traditional cultural trading services are only one aspect of how digital technology affects the transformation and upgrading of the cultural industry. Another example is the effective matching of consumers and products created by the various channels created by digital technology and internet architecture. Digital production techniques, sales channels, and feedback systems enable the provision of a varied and precise product supply. In order to enhance the cultural industry value chain, digital technology is used in conjunction with the provision of traditional cultural products and services. With the further development of digital transformation, enterprises can achieve fine management throughout the product life cycle, eliminate inefficient production capacity, lower production costs, and increase production efficiency. For industries, close

internal resource integration, resource network advantages, improved dynamic response capabilities, and a better ability to adapt to the constantly changing cultural consumption needs can be achieved through digital technology.

Secondly, unlike ordinary consumer goods, the market demand for cultural products is highly variable and closely related to consumers' spending power. As a result, the expected revenue of cultural companies is subject to greater uncertainty. For example, it is often more difficult to forecast movie ticket sales and bestselling book releases than ordinary physical goods. In addition, cultural products often have positive externalities and product features can be easily replicated, which can also increase the volatility of cultural enterprises' revenues; therefore, this can lead to greater investment risks for financial institutions. In the digital economy, digital technology can replace most manual activities, integrate online and offline resources scientifically and efficiently, broaden the source of information collection, collect the flow data of relevant cultural enterprises in real-time, dynamically track and analyze investment risks, and make corresponding adjustments in a timely manner, thus enhancing the risk pricing ability and risk control ability and reducing the cost of financing services for the cultural industry [24].

In conclusion, the digital economy has significantly decreased the degree of information asymmetry between investment and financing, lowered the financing costs of cultural enterprises, provided financial support for their R&D investment and transformation of achievements, and effectively promoted sustainable development in cultural industries through the use of contemporary digital tools, like big data and cloud computing [25]. The following hypothesis is put out in light of the analyses just mentioned:

**Hypothesis 1.** *The digital economy significantly and favorably affects sustainable development in the cultural sector.*

*3.2. The Moderating Effect of Government Support*

The government-led governance paradigm, which is unusual compared to Western nations, is essential to China's cultural industry's sustainable development. The cultural sector is a developing one with its own distinct laws of development. The institutional structure is not yet flawless, and the general development of China's cultural industry is still in its early phases. Therefore, through institutional development and policy assistance, the government offers the necessary direction and support, fostering an environment favorable to that industry's sustainability [26].

First, the government can study the trends that are expected to shape the development of the cultural sector, adopt pertinent policies to support those predictions, effectively advocate for the sector's sustainable growth, release relevant development reports and strategic plans in a timely manner, and clarify development trends and priorities [27]. Second, government assistance can serve to foster the fusion of cultural components with the real economy, increase the economic added value of associated businesses, and create a chain of industrial processes that starts with the "development of cultural resources and ends with the transformation of cultural components" [28]. This will increase the availability of effective high-quality cultural content, widen the channels of dissemination, and promote the value of good traditional Chinese culture today. Again, the construction of a good culture industry ecology cannot be achieved without government input and policy support [29]. The government has created a relaxed business environment for sustainable development in cultural industries through its ability to optimize the structural layout of cultural industries and improve the infrastructure, laws, and regulations needed for their sustainable development. This boosts the momentum of broad invention and creativity in the cultural sector, alters the sector's initial development trend, and creates new industries, improving the sector's overall strength and competitiveness of development [30]. Therefore, the likelihood that the digital economy will encourage sustainable development

in the cultural industry increases with the level of government support for the sector. The following theory is put out in light of the analyses just mentioned:

**Hypothesis 2.** *The process of the digital economy affecting sustainable development in the cultural industries is moderated by government support.*

### 4. Methods

*4.1. Variable Selection and Data Sources*

4.1.1. Dependent Variable: Cultural Industry Development Level (CUL)

Most existing studies believe that the sustainable development of the cultural industry should include four dimensions: innovation, coordination, openness, and sharing. In order to reflect the multiple attributes of sustainable development, we consider the particularity, the quantifiability of indicators, and the availability of data. Drawing on Li et al. (2023), this paper constructs a comprehensive indicator system to measure the sustainable development level of the cultural industry from dimensions such as cultural innovation ability, coordination level, openness level, sharing level, and industrial efficiency [31]. The indicator data in the indicator system mainly comes from the National Research Network database, the Statistical Yearbook of Culture and Related Industries, and the Guotai An database; partial missing data are calculated using trend prediction or interpolation methods. Finally, a principal component analysis was conducted on the five indicators of cultural innovation ability, collaboration level, openness, sharing level, and industrial efficiency, ultimately obtaining a comprehensive indicator. The details are shown in Table 1.

**Table 1.** Indicator system for measuring the sustainable development level of the cultural industry.

| Primary Indicators | Secondary Indicators | Properties |
|---|---|---|
| Innovation capability | Number of cultures, art, science and technology, and scientific research institutions | + |
| | Culture, art, science and technology, and research institutions' assets | + |
| | Number of professional and technical personnel in cultural research institutions | + |
| | R&D investment intensity | + |
| | Number of patents obtained by cultural enterprises | + |
| | Number of works copyrighted by cultural enterprises | + |
| | Number of software copyrights obtained by cultural enterprises | + |
| Coordination level | Ratio of per capita cultural consumption to total consumption | + |
| | Advanced industrial structure | + |
| Degree of openness | Number of participants in foreign cultural exchange activities | + |
| | Number of foreign cultural exchange projects | + |
| Sharing level | Public library holdings per capita | + |
| | Public library floor space per capita | + |
| | Museum collections per 10,000 people | + |
| Industry benefits | Above-average cultural manufacturing, wholesale, and services' business taxes and surcharges | + |
| | Museum visits | + |
| | Number of people attending lectures at mass cultural institutions and libraries | + |

### 4.1.2. Independent Variable: Digital Economy (DIG)

Drawing on Huang et al. (2019), this paper measures the level of development of the digital economy in each region from the perspectives of both internet development and digital inclusive financial development [32]. To measure the development of the internet, this paper draws on the method of Huang Huiqun et al. (2019) by choosing the penetration rate of mobile phones, the employment situation in the information industry, the output situation related to the internet, and the penetration rate of the internet. To measure the development of digital financial inclusion in China, choose the digital inclusion index and weigh the five indicators collectively to determine the level of urban and regional digital inclusion.

### 4.1.3. Moderating Variable: Government Support (GOV)

Drawing on Wall-Andrew et al. (2021), we chose the share of expenditure on culture, sports, and media in the general public budget expenditure to measure the degree of government support for development in cultural industries [33].

### 4.1.4. Control Variables

In order to avoid other factors affecting development in cultural industries from interfering with our empirical results, Drawing on Liu et al. (2022), we control for the following variables [34]: (1) per capita gross national product (Pgdp); (2) financial institutions' deposit and loan balances as a percentage of the province's gross domestic product (GDP) in that year (Finance), calculated as the ratio of the institutions' deposit and loan balances to the GDP level of the province; (3) general public budget expenditure on culture (Budgcul); (4) culture industry endowment structure (Bingfu), as determined by the capital stock to labor force participation ratio in businesses larger than the industry; (5) the number of employees in cultural institutions (Shiye); (6) the average number of students in higher education (Educ). Finally, we controlled for province and year-fixed effects. Data for each control variable were obtained from various public information sources such as the China Statistical Yearbook, the China Information Industry Yearbook, and the database of the National Bureau of Statistics.

### 4.2. Sampling Technique

We analyzed the panel data created with 30 provinces in China within a sample period of 2009–2020. In order to avoid bias in the data, we performed the following with the original sample data:

(1) In order to avoid bias in the estimation results due to the existence of extreme values, all the variables are shrink-tailed at the 1% level.
(2) Partial missing data ar calculated using trend prediction or interpolation methods.

This article constructs a comprehensive indicator system to measure the sustainable development level of the cultural industry from dimensions such as cultural innovation ability, coordination level, openness level, sharing level, and industrial efficiency. The indicator data in the indicator system mainly come from the National Research Network database, the Statistical Yearbook of Culture and Related Industries, and the Guotai An database; partial missing data are calculated using trend prediction or interpolation methods. Finally, a principal component analysis was conducted on the five indicators of cultural innovation ability, collaboration level, openness, sharing level, and industrial efficiency, ultimately obtaining a comprehensive indicator. In addition, data for the independent variable digital economy were obtained from the Digital Financial Inclusion Index and the China Statistical Yearbook. Data for the moderating variable, government support, and the control variables were obtained from the China Statistical Yearbook, the China Information Industry Yearbook, and the National Bureau of Statistics database.

### 4.3. Econometric Approach

This article builds two empirical models using data from 2009 to 2020. The first model examines how the digital economy affects the cultural sector's ability to grow sustainably. This paper focuses on the coefficient $\alpha_1$ of $DIG_{it}$, which reflects the impact of the digital economy on the sustainable development of the cultural industry. $\alpha_1$ indicates the degree of impact of the digital economy on the sustainable development of the cultural industry.

$$CUL_{it} = \alpha_0 + \alpha_1 DIG_{it} + \alpha_2 GOV_{it} + \alpha_3 X_{it} + \xi_i + \eta_t + \mu_{it}$$

where $i$ denotes the province and $t$ denotes the year. $CUL_{it}$ denotes the level of development in the cultural industry in province $i$ in year $t$. $DIG_{it}$ denotes the level of development in the digital economy in province $i$ in year $t$. $GOV_{it}$ denotes the government support for development in the cultural industry in year $t$ in province $i$. $X_{it}$ denotes a set of control variables. $\xi_i$ represents the province dummy variable. $\eta_t$ represents the time dummy variable. $\mu_{it}$ is the random disturbance term. To increase the accuracy of the regression results, we treated all of the variables as logarithms.

The second model examines how government support may mitigate the impact of the digital economy on the long-term growth of the cultural industries. This paper focuses on the interaction $DIG_{it} \times GOV_{it}$ between the digital economy and government support. $DIG_{it} \times GOV_{it}$ is the interaction term between the level of digital economy development and government support for the cultural industry, and its coefficient. $\beta_3$ denotes the extent to which the impact of the digital economy on development in the cultural industry is moderated by government support.

$$CUL_{it} = \beta_0 + \beta_1 DIG_{it} + \beta_2 GOV_{it} + \beta_3 DIG_{it} \times GOV_{it} + \beta_4 X_{it} + \xi_i + \eta_t + \mu_{it}$$

where $i$ denotes the province and $t$ denotes the year. $CUL_{it}$ denotes the level of development in the cultural industry in province $i$ in year $t$. $DIG_{it}$ denotes the level of development in the digital economy in province $i$ in year $t$. $GOV_{it}$ denotes the government support for development in the cultural industry in year $t$ in province $i$. $DIG_{it} \times GOV_{it}$ is the interaction term between the level of digital economy development and government support for the cultural industry and its coefficient. $X_{it}$ denotes a set of control variables. $\xi_i$ represents the province dummy variable. $\eta_t$ represents the time dummy variable. $\mu_{it}$ is the random disturbance term. To increase the accuracy of the regression results, we treated all of the variables as logarithms.

### 4.4. Empirical Analysis

#### 4.4.1. Model Checking

We performed normality, heteroskedasticity, autocorrelation, and covariance tests before performing OLS regression to ensure that our data fit the OLS model.

A normality test is when the values of the predictor variables are fixed and the dependent variable is normally distributed, the residual values should also be a normal distribution with a mean of zero. To test for normality, we plotted the standard residual histogram and the normal P-P plot. As shown in Figure 1, the standardized residuals of this regression approximated a normal distribution. Also, as shown in Figure 2, the points in the normal P-P plot fall essentially on the diagonal line, indicating that the regression residuals are close to a normal distribution.

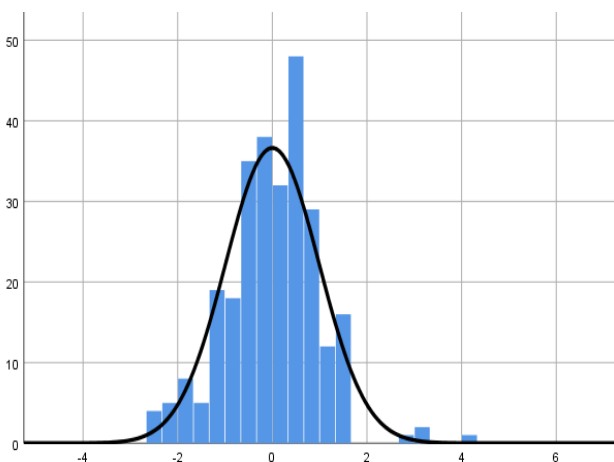

**Figure 1.** Histogram of regression-standardized residuals.

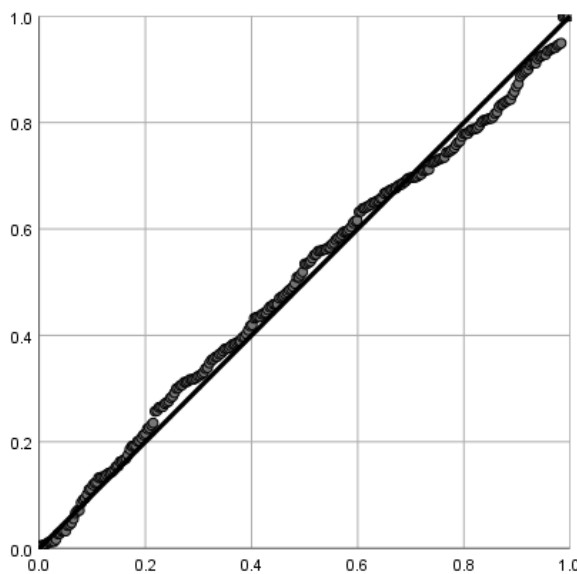

**Figure 2.** Normal P-P diagram of regression-standardized residuals.

For the heteroskedasticity test, we used the Breusch–Pagan/Cook–Weisberg test. The results show a *p*-value of 0.7879 > 0.05; so the original hypothesis (Ho: constant variance) is not rejected, indicating that there is no heteroskedasticity problem in the data.

For the autocorrelation test, we used the lagrange multiplier test (LM test) for autocorrelation in panel data, and the result shows that the *p*-value is 0.0963 > 0.05; so the original hypothesis of the LM test (Ho: no autocorrelation) is not rejected, indicating that there is no autocorrelation problem in the data. Also, it shows that the OLS model is more appropriate for the sample data than the random effects model.

For the covariance test, to ensure that multicollinearity did not affect the results, we calculated variance inflation factors (VIFs). The results showed that the VIFs for all variables were below 6.03 (mean = 3.01), well below the generally accepted threshold of 10.0. We also tested, as seen in Table 2, the correlation coefficients between the variables, which were mostly less than 0.70, which is the lowest limit for which multicollinearity is considered possible. This indicates that the covariance problem of the data in this paper is small.

**Table 2.** Descriptive statistics and correlation analysis of variables.

| Variables | Obs | Mean | S. D. | (1) | (2) | (3) | (4) | (5) | (6) | (7) | (8) | (9) |
|---|---|---|---|---|---|---|---|---|---|---|---|---|
| (1) CUL | 298 | 3.579 | 2.267 | 1.000 | | | | | | | | |
| (2) DIG | 298 | 1.464 | 0.443 | 0.385 | 1.000 | | | | | | | |
| (3) Pgdp | 298 | 4.049 | 3.329 | 0.362 | 0.300 | 1.000 | | | | | | |
| (4) Finance | 298 | 3.822 | 1.24 | 0.469 | 0.208 | 0.078 | 1.000 | | | | | |
| (5) Budgcul | 298 | 0.987 | 0.115 | −0.489 | −0.012 | −0.090 | 0.193 | 1.000 | | | | |
| (6) Bingfu | 298 | 0.916 | 2.278 | −0.528 | 0.065 | −0.011 | 0.208 | −0.045 | 1.000 | | | |
| (7) Shiye | 298 | 1.097 | 0.297 | 0.268 | −0.043 | 0.013 | 0.034 | 0.040 | −0.166 | 1.000 | | |
| (8) Educ | 298 | 3.413 | 1.753 | −0.592 | −0.004 | −0.300 | −0.027 | −0.062 | 0.173 | −0.108 | 1.000 | |
| (9) GOV | 298 | 1.581 | 0.494 | −0.531 | 0.111 | −0.048 | 0.039 | 0.010 | 0.224 | 0.064 | −0.086 | 1.000 |

Using the above four tests, we believe that our data meet the basic conditions of OLS regression. Therefore, we choose the mixed-effects model, i.e., the POLS model.

### 4.4.2. Descriptive Statistics

Descriptive statistical studies were performed on the variables to ensure that multicollinearity did not alter the results. The results show that the maximum correlation coefficient between all variables is 0.592, which is less than 0.70, which is the minimum required to show likely multicollinearity. Table 2 displays the statistical information for the specified sample.

### 4.4.3. Baseline Regression Analysis

To test our research hypotheses, we employed an OLS model. Table 3 displays the OLS regression results of the impact of the digital economy on cultural industry sustainable development. First, Model 1 evaluates the impact of the digital economy on sustainable development in cultural industries without taking into account the control factors; then, in Models 2–7, we incorporated our control variables sequentially, always controlling for variances related to province and time. The findings reveal that with and without controlling variables, the digital economy regularly and positively contributes to sustainable development in cultural business. The findings support our contention that the digital economy can speed the flow of innovative variables such as knowledge, technology, and capital to enhance development in cultural industries, and thus H1 is supported.

**Table 3.** Baseline regression results.

| Variables | Model 1 | Model 2 | Model 3 | Model 4 | Model 5 | Model 6 | Model 7 |
|---|---|---|---|---|---|---|---|
| | CUL | CUL | CUL | CUL | CUL | CUL | CUL |
| DIG | 0.170 *** | 0.073 ** | 0.106 *** | 0.053 ** | 0.027 *** | 0.057 *** | 0.096 *** |
| | (0.012) | (0.049) | (0.020) | (0.017) | (0.043) | (0.028) | (0.016) |
| Pgdp | | 0.032 *** | 0.035 *** | 0.118 *** | 0.190 *** | 0.046 *** | 0.039 *** |
| | | (0.102) | (0.162) | (0.149) | (0.157) | (0.053) | (0.063) |
| Finance | | | 0.106 ** | 0.062 * | 0.107 ** | 0.074 *** | 0.085 *** |
| | | | (0.126) | (0.349) | (0.647) | (0.235) | (0.404) |
| Budgcul | | | | 0.409 ** | −0.234 *** | −0.311 ** | 0.181 *** |
| | | | | (0.888) | (0.639) | (0.983) | (0.047) |
| Bingfu | | | | | 0.304 * | 0.208 *** | −0.783 *** |
| | | | | | (2.963) | (2.997) | (0.655) |
| Shiye | | | | | | −0.471 * | −0.374 *** |
| | | | | | | (2.634) | (0.59) |
| Educ | | | | | | | −0.078 |
| | | | | | | | (0.621) |
| Constants | 0.588 ** | 0.043 * | 0.964 * | 0.637 *** | 0.515 *** | 0.488 *** | 0.095 ** |
| | (0.234) | (0.17) | (0.193) | (0.553) | (0.006) | (0.102) | (0.185) |

**Table 3.** *Cont.*

| Variables | Model 1 | Model 2 | Model 3 | Model 4 | Model 5 | Model 6 | Model 7 |
|---|---|---|---|---|---|---|---|
| | CUL | CUL | CUL | CUL | CUL | CUL | CUL |
| Year dummy | yes | yes | yes | yes | yes | yes | yes |
| Province dummy | yes | yes | yes | yes | yes | yes | yes |
| Observations | 298 | 298 | 298 | 298 | 298 | 298 | 298 |
| R-squared | 0.303 | 0.465 | 0.782 | 0.799 | 0.806 | 0.862 | 0.908 |

Note: Standard errors are in parentheses; *** $p < 0.01$, ** $p < 0.05$, * $p < 0.1$.

### 4.4.4. Regression Analysis of Moderating Effects

The regression findings for the moderating effect of government support on the influence of the digital economy and the level of development in cultural industries are shown in Table 4. Model 1 contains only the results of the baseline model with the control variables. In Model 2, we include the control variable, the independent variable, and the digital economy. In Model 3, we add the control variable, the independent variable, the digital economy, the moderator variable government support, and the interaction term between the independent variable and the moderator variable to test the moderating effect of government support in the process of sustainable development of the cultural industry influenced by the digital economy. Model 3 reveals that the coefficient of DIG*GOV is 0.102, indicating that the promotion effect of the digital economy on development in the cultural industry is further enhanced by strong government policy support, i.e., government support plays a positive moderating role in the relationship between the digital economy and the level of development in the cultural industry. Hypothesis two was supported.

**Table 4.** Regression results of moderation effects.

| Variables | Model 1 | Model 2 | Model 3 |
|---|---|---|---|
| | CUL | CUL | CUL |
| DIG | | 0.096 *** | 0.102 ** |
| | | (0.016) | (0.012) |
| DIG × GOV | | | 0.102 ** |
| | | | (0.053) |
| GOV | | | 0.032 |
| | | | (0.305) |
| Pgdp | 0.023 *** | 0.039 *** | 0.489 *** |
| | (0.361) | (0.063) | (0.054) |
| Finance | 0.053 *** | 0.085 *** | 0.965 * |
| | (0.421) | (0.404) | (0.476) |
| Budgcul | 0.336 ** | 0.181 *** | 0.221* |
| | (0.084) | (0.047) | (0.556) |
| Bingfu | 0.402 ** | −0.783 *** | 0.181 *** |
| | (0.784) | (0.655) | (0.104) |
| Shiye | 0.496 *** | 0.374 *** | 0.763 |
| | (0.671) | (0.59) | (0.059) |
| Educ | −0.227 * | −0.078 | −0.015 |
| | (0.405) | (0.621) | (0.321) |
| Constants | 0.032 * | 0.095 ** | 0.143 ** |
| | (0.298) | (0.185) | (0.867) |
| Year dummy | yes | yes | yes |
| Province dummy | yes | yes | yes |
| Observations | 298 | 298 | 298 |
| R-squared | 0.893 | 0.908 | 0.902 |

Note: Standard errors are in parentheses; *** $p < 0.01$, ** $p < 0.05$, * $p < 0.1$.

### 4.4.5. Robustness Regression Results

To ensure the reliability of the estimation results, this paper changes the measurement of the dependent variable. This paper uses the total output of cultural industries per capita (the ratio of the total output of local cultural industries to the number of the population in that year) to measure the development of cultural industries in each province as the explanatory variable in the empirical analysis. The data are calculated based on relevant data from the China Culture and Cultural Relics Statistical Yearbook and the China Statistical Yearbook. Tables 5 and 6 show the regression results of the robustness tests. Table 5 shows the impact of the digital economy on the sustainability of the cultural industry. As can be seen in Table 5, the DIG coefficient is 0.279, indicating that the digital economy has a positive and significant impact on the sustainable development of the cultural industry. Hypothesis one is again validated.

**Table 5.** Baseline regression results.

| Variables | Model 1 | Model 2 | Model 3 | Model 4 | Model 5 | Model 6 | Model 7 |
|---|---|---|---|---|---|---|---|
| | CUL | CUL | CUL | CUL | CUL | CUL | CUL |
| DIG | 0.279 *** | 0.271 *** | 0.283 *** | 0.280 *** | 0.236 *** | 0.230 *** | 0.279 *** |
| | (0.039) | (0.049) | (0.051) | (0.050) | (0.051) | (0.052) | (0.050) |
| Pgdp | | 0.081 *** | 0.081 *** | 0.078 *** | 0.125 *** | 0.130 *** | 0.079 *** |
| | | (0.024) | (0.023) | (0.022) | (0.026) | (0.026) | (0.026) |
| Finance | | | −0.068 ** | −0.058 ** | −0.020 | −0.020 | −0.054 ** |
| | | | (0.027) | (0.027) | (0.028) | (0.028) | (0.026) |
| Budgcul | | | | −2.112 *** | −2.191 *** | −2.130 *** | −2.316 *** |
| | | | | (0.548) | (0.495) | (0.493) | (0.815) |
| Bingfu | | | | | −1.387 *** | −1.466 *** | −1.393 *** |
| | | | | | (0.267) | (0.279) | (0.298) |
| Shiye | | | | | | −0.525 * | −0.526 * |
| | | | | | | (0.294) | (0.295) |
| Educ | | | | | | | −0.699 *** |
| | | | | | | | (0.130) |
| Constants | 1.87 *** | −1.857 | −1.318 | 0.873 | −0.378 | −0.525 | 4.663 *** |
| | (0.175) | (1.204) | (1.189) | (1.339) | (1.527) | (1.543) | (1.793) |
| Year dummy | yes | yes | yes | yes | yes | yes | yes |
| Province dummy | yes | yes | yes | yes | yes | yes | yes |
| Observations | 461 | 324 | 324 | 324 | 316 | 316 | 298 |
| R-squared | 0.131 | 0.217 | 0.230 | 0.239 | 0.288 | 0.291 | 0.348 |

Note: Standard errors are in parentheses; *** $p < 0.01$, ** $p < 0.05$, * $p < 0.1$.

Table 6 shows the moderating effect of government support on the digital economy affecting the sustainable development of the cultural industry. As can be seen in Table 6, the DIG × GOV coefficient is 2.438, indicating that the promotion effect of the digital economy on the development of the cultural industry is further enhanced by the strong support of government policies, i.e., government support plays a positive moderating role in the relationship between the digital economy and the development level of the cultural industry. Hypothesis two is again validated.

**Table 6.** Regression results of moderation effects.

| Variables | Model 1 | Model 2 | Model 3 |
|---|---|---|---|
| | CUL | CUL | CUL |
| DIG | | 0.279 *** | 0.328 *** |
| | | (0.050) | (0.050) |
| DIG × GOV | | | 2.438 *** |
| | | | (0.655) |
| GOV | | | −1.138 *** |
| | | | (0.274) |
| Pgdp | 0.185 *** | 0.079 *** | 0.049 * |
| | (0.025) | (0.026) | (0.027) |
| Finance | 0.003 | −0.054 ** | −0.079 *** |
| | (0.028) | (0.026) | (0.027) |
| Budgcul | −2.170 *** | −2.316 *** | −1.884 *** |
| | (0.586) | (0.815) | (0.725) |
| Bingfu | −0.774 ** | −1.393 *** | −0.916 ** |
| | (0.307) | (0.298) | (0.373) |
| Shiye | −0.627 * | −0.526 * | −0.266 |
| | (0.355) | (0.295) | (0.373) |
| Educ | −0.446 *** | −0.699 *** | −0.870 *** |
| | (0.153) | (0.130) | (0.141) |
| Constants | −1.666 | 4.663 *** | 4.143 ** |
| | (1.762) | (1.793) | (1.867) |
| Year dummy | yes | yes | yes |
| Province dummy | yes | yes | yes |
| Observations | 343 | 298 | 298 |
| R-squared | 0.227 | 0.348 | 0.394 |

Note: standard errors are in parentheses; *** $p < 0.01$, ** $p < 0.05$, * $p < 0.1$.

## 5. Discussion and Conclusions

Using the above theoretical deduction and empirical tests, this paper finds that the digital economy has a substantial role in promoting the sustainable development of the cultural industry. The digital economy of the cultural industry has transformed the entire industry and industrial chain of the traditional cultural industry through digital technology, reflecting the multiplier and superposition impact of digital technology on promoting the development of the cultural industry. Big data, artificial intelligence, cloud computing, and the other digital economy, data analysis, and the accurate management of the production, communication, consumption, and other aspects of the cultural industry are used to achieve a better user experience, a higher market share, better economic benefits, and further realize the digital economy to promote the sustainable development of the cultural industry. In addition, the favorable impact of the digital economy on the growth of the cultural industry has been further enhanced with the strong support of the government.

The development mode of the digital economy has become an important guarantee for maintaining economic development and an important means to promote the sustainable development of the cultural industry. To this end, the goal of promoting the sustainable development of the cultural industry by the digital economy should be clearly defined to maximize the efficiency of the digital economy. First of all, improve the management level of the cultural industry, increase the compliance construction of the cultural industry, and improve the attractiveness of the cultural industry to the digital economy. Second, strengthen the research and development of digital technology in the era of the digital economy, improve the development level of the digital economy, and implement the effect of digital technology to empower the cultural industry. Third, promote the integrated development of the digital economy and the cultural industry to achieve a positive interaction between the digital economy and the sustainable development of the cultural industry. Fourth, improve the supporting system for the sustainable development of the digital economy and the cultural industry, improve the regulatory system of the digital

economy, and improve the governance capacity of the cultural industry. Fifth, pay attention to the cultivation of interdisciplinary talents in the field of cultural industry under the background of the digital economy and make full use of human resources to give full play to the subjective initiative of integrated development.

Based on the above research results and the goal of the digital economy promoting the sustainable development of the cultural industry, this paper puts forward the following policy suggestions for the sustainable development of China's cultural industry in the era of digital economy.

### 5.1. Strengthen the Establishment and Improvement of the Talent Training System of the Digital Culture Industry

First of all, a talent demand management system and recruitment mechanism should be set up first by the relevant departments of the digital culture sector [35]. Active regulation policies should be implemented by the relevant departments in accordance with the specific talent requirements for the entire industrial chain, and training programs and introduction structures for various levels and types of talent, such as those in digital economy technology, digital management technology, and digital operation technology, should be timely adjusted.

Second, digital technology education should be fully integrated into the ensuing educational program, and different companies should be used as teaching platforms to develop fundamental skills in line with the demands of the cultural sector. Universities and professional research institutions should simultaneously act as bridges by consciously cultivating top-tier research talent to advance the ground-breaking advancement of digital technology [36]. We can fully satisfy the demands of various businesses for digital talent by enhancing the cultural industry's professional and educational levels in both areas.

Finally, in order to fully realize the subjective initiative of integrated development, focus should be given to fostering compound talent. The development of composite talent should be a priority for all levels of government, and initiatives to build composite-leading talent projects and proactively introduce composite talent should be supported. On the other hand, the cultural and digital finance sectors should also actively develop and utilize the already-available professional talent resources, improve employees' industry literacy through on-the-job training, etc., and transform single-type talent into composite talent to ensure that the digital economy contributes to cultural development that is sustainable [37].

### 5.2. Scientifically Plan the Digital Development Strategy of the Cultural Industry

First, support original thinking. Create an independent innovation-driven digital cultural industry with diversified channel development as the direction, and actively cultivate high-end creative cultural industries. We will assist businesses in enhancing their capacity for independent innovation, aid various high-tech sectors in fully integrating with the cultural sector, and systematically improve the innovation capacity of the cultural sector.

Second, the development of cultural industry clusters must also be encouraged. In order to create industrial clusters of a certain size, we should actively introduce and integrate a variety of supporting industries that cater to the needs of the digital development of the cultural industry. We should also take a number of effective steps to actively support businesses with growth potential so they can become leading businesses [38]. Additionally, by utilizing the agglomeration effect, we can ensure the establishment and advancement of a public digital technology platform for the cultural industry, create a pathway for the quick growth of industrial clusters, and successfully accomplish the objective of intensive industrial development [39].

Thirdly, the brand effect must be utilized. To ensure that the cultural enterprises, cultural brands, and service brands developed have independent research and development capabilities and, more importantly, independent intellectual property rights, we should be committed to strengthening the close cooperation between high-tech enterprises and

creative design enterprises [40]. This will enable us to provide dynamic support for the ongoing development of corporate brands in the digital cultural industry base.

Fourthly, in order to encourage talent development, a comprehensive talent training program must be established. The digital talent cultivation of the cultural industry needs to fully incorporate contemporary educational ideas and a variety of approaches. We also need to build an international platform with effective channels for information sharing for cultural industry talent. Based on this, actively cultivate the required management, compound, and cultural talents, and provide long-term talent support for the sustainable development of China's digital culture industry.

Fifth, create a business ecosystem to encourage theoretical and practical innovation throughout the entire industrial chain, make effective use of digital infrastructure, and raise the industrial chain's stability and competitiveness [41]. The legislative environment should be improved, new businesses and business models should be examined with an open mind, varied innovations should be supported, and the digital cultural sector should be given enough room to grow while carefully sticking to the principle of safe development [36].

### 5.3. Promote the Development of High-Quality Digital Cultural Products

The main element in enhancing industry development is product quality. The government has to understand how the cultural sector is evolving and enhance its supply chain [42]. Virtualization and digital transformation are the future of cultural products, according to modern cultural industry development practices [43]. In keeping with China's strategic goal of green growth, virtualization and the digital production of cultural products can reduce the expenses of both time and space while also improving the effectiveness of resource usage. Future cultural industry value chains will be led by intelligent and digital products that meet needs and provide value through constant user involvement [44]. China has made progress in the sustainable development of digital cultural items in recent years. Ant, Tencent, and Byte Jump are just a few of the major internet goliaths that have started similar enterprises. Since the "Treasure Plan" was introduced, approximately 20 institutions have subscribed to it and released "digital collections," for which "Ant Chain" has provided technological support. The creation of digital cultural products offers fresh perspectives for the sustainable growth of the cultural industry, particularly in the context of global cultural businesses pursuing green transformation.

### 5.4. Continuing to Build a Modern Cultural Market System

Innovation and creativity are the driving force and source of growth in the digital culture sector, and initiatives to foster these traits must respect and uphold citizens' fundamental cultural rights, foster their innovative capacity and creativity, and maintain a calm and orderly market environment. In order to address the unresolved issues in the process of performing the function of maintaining market order, the government should support the creation of legislation and the standardization of law enforcement, dismantle the current system of managing the cultural market, which relies excessively on administrative tools and other issues, define the boundaries between the government and the market, and effectively safeguard citizens' cultural rights and the environment for industrial development. The digital culture industry is closely related to developments in high technology, and the characteristics of this industry development highlight the fact that enterprises and industry associations are more familiar with and can master the current situation and laws of the industry development better than the competent government departments and are thus suitable to participate in the management of the industry development. Due to this, it is essential to fully exploit the enthusiasm and initiative of businesses and industry associations, bring together social forces under the direction of the government, create a modern governance system with Chinese characteristics, strengthen the governance capacity of the government, and create the best possible development environment for the cultural industry.

**Author Contributions:** R.W. participated in the conceptualization and design of the study. R.S., K.Z. and X.W. carried out the review of the literature. All of the contributors provided feedback on earlier drafts of the book after Q.W. wrote the original draft. All authors have read and agreed to the published version of the manuscript.

**Funding:** The National Social Science Fund Key Project of China (20AZD095) provided funding for this endeavor.

**Informed Consent Statement:** All individuals taking part in the study gave their informed consent.

**Data Availability Statement:** The data and models used during the study are available from the corresponding author by request.

**Conflicts of Interest:** The authors declare no conflict of interest.

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
