# Peer review of "How the Digital Economy Enables Regional Sustainable Development Using Big Data Analytics"

_sustainability, doi:10.3390/su151813610_

Round 1

Reviewer 1 Report (Previous Reviewer 2)

1.       Include the method and originality of this study in the abstract.

2.       The scenario gaps of the digital economy and Big Data Analytics shall be discussed, and clearly state which part of the gap the authors can fill in this study.

3.       Compare the China’s cultural industry and its initiative for sustainable development to other countries. Why do we need to study in the China context?

4.       Include the discussion on the selection variables to be included in the model.

5.       Change the methodology section into methods. Include the procedure of data collection, sampling techniques and analysis.

6.       Include the notation from the formula and explain how authors represent a function rule in function notation.

7.       The authors need to include the classical assumptions of multiple regression.

8.       How did the authors deal with the data bias?

9.       Theoretical and practical implications shall include the discussion of sustainable development and its goals.

Check the complex sentences and logical connections from one paragraph to another. 

Author Response

Response Letter to the Comments of Reviewer #1

“How the Digital Economy Enables Regional Sustainable Development using Big Data Analytics”

Submitted to

Sustainability

Dear Reviewer #1,

We would like to thank you for the comments to this paper. We hope that you find it considerably improved. In the revision, we have strived as much as possible to address all the issues raised by you. Below, we provide detailed responses to your comments (We have highlighted the changes in red in the article).

  1. Include the method and originality of this study in the abstract.

Response to Comment 1: 

Many thanks for pointing out to this important weakness. We add in the abstract section the method used in this paper, the model, and the innovations of this paper. The details are as follows:

Abstract: The development of the cultural industry cannot be isolated from the efficient integration with the digital economy and digital technology at the current stage of the technological and industrial revolution. This paper constructs an indicator system to measure the sustainable development of the cultural industry and tests the relationship between the digital economy and the sustainable development of the cultural industry using an OLS model based on China's provincial panel data from 2011 to 2021. The findings of the study suggest that the digital economy can significantly aid in the long-term growth of cultural companies. The process of promoting sustainable development of the cultural industry through the digital economy has also advanced thanks to the government’s strong support. This report also suggests governmental recommendations based on these findings for the sustainable development of China’s cultural industry in the age of the digital economy. This paper theoretically elucidates the mechanism of the role of digital economy on the sustainable development of the cultural industry; constructs a system of indicators to measure the sustainable development of the cultural industry, and tests the impact of the digital economy on the sustainable development of the cultural industry.

  1. The scenario gaps of the digital economy and Big Data Analytics shall be discussed, and clearly state which part of the gap the authors can fill in this study.

Response to Comment 2: 

Many thanks for pointing out to this important weakness. We have added the current status of research on sustainable development of cultural industry in the introduction and clarified the research gaps of existing research on sustainable development of cultural industry, and this paper will explore the impact on sustainable development of cultural industry from the perspective of digital economy to fill some of the theoretical gaps. The specific contents are as follows:

The national cultural system is carried by the cultural industry, which is a significant part of the national economic industrial system. Promoting the cultural sector’s sustainable growth serves two purposes: it fosters high-quality economic growth and increases cultural pride and self-improvement. The “China Cultural Industry Investment and Financing Report (2021)” demonstrates that, although the general investment and financing position of China’s culture industry is improving, there are still significant problems, including uneven socioeconomic gains and shoddy industrial investment. In the process of transforming the digital economy from consumption to production, the penetration of new technologies has created new forms and models of cultural consumption, and has become a new driving force for the sustainable development of the cultural industry. Therefore, it is of significant theoretical value to describe the boosting mechanism of digital economy in the sustainable development of cultural sector given the general trend of rapid development of digital technologies such as blockchain, Big Data, and 5G network. In addition, existing studies have paid extensive attention to the impact of cultural innovation [5], talent cultivation [6], branding [7], industrial integration [8] and other factors on the sustainable development of the cultural industry. While existing research provides important insights into how to promote the sustainable development of cultural industries, few scholars have explored the impact on the sustainable development of cultural industries from the perspective of the digital economy and big data analytics. The next section of the research will analyze how the digital economy supports the sustainable development of the cultural sector and look for workable implementation strategies to achieve that sector’s sustainable development.

[5]Yan, W. J., & Liu, S. T. (2023). Creative Economy and Sustainable Development: Shaping Flexible Cultural Governance Model for Creativity. Sustainability, 15(5), 4353.

[6]Jin, W., Gao, S., & Pan, S. (2023). Research on the impact mechanism of environmental regulation on green total factor productivity from the perspective of innovative human capital. Environmental Science and Pollution Research, 30(1), 352-370.

[7]Kádár, B., & Klaniczay, J. (2022). Branding Built Heritage through Cultural Urban Festivals: An Instagram Analysis Related to Sustainable Co-Creation, in Budapest. Sustainability, 14(9), 5020.

[8]Adamik, A., & Sikora-Fernandez, D. (2021). Smart organizations as a source of competitiveness and sustainable development in the age of industry 4.0: Integration of micro and macro perspective. Energies, 14(6), 1572.

  1. Compare the China’s cultural industry and its initiative for sustainable development to other countries. Why do we need to study in the China context?

Response to Comment 3: 

Many thanks for pointing out to this important weakness. We have added a literature review section, in which we compare the sustainable development of China's cultural industry with other countries. It also clarifies why it is important to conduct the study in the Chinese context. The details are as follows:

  1. Literature review

The cultural industry is a global priority for all nations [9], but in contrast to developed nations, China still lags far behind in the five areas of capital investment, production efficiency, industrial scale, policies and regulations, and technological innovation [10]. China’s current policy system for the cultural industry is still not perfect enough, and the targeting and operability are not strong [11]. In contrast, the United States, South Korea, and other nations regulate and promote the development of the cultural industry through legislation and other measures, and have formed a complete legal system in the protection of intellectual property rights, movie grading and rating system, etc [12]. Furthermore, China’s cultural industry has developed more quickly than that of developed nations, which prevents it from achieving the inclusive development of cultural plurality [13]. Additionally, given how digital technology has affected China’s traditional cultural industry, China’s digital cultural industry has a greater need for institutional innovation in order to support the long-term production of cultural creativity in China’s cultural industry [14].

Because China’s cultural sector is undergoing a digital transformation, it serves as an excellent case study for the noteworthy characteristics of the new digital cultural sector, including significant clustering, connectivity [15], virtualization [16], and challenges related to humanism and ethics [17], cultural experience differences [18], as well as regional and spatial organization. The growth of China’s cultural industry offers numerous examples and rich practices for the transformation and study of the global cultural industry [19]. Therefore, we will take the sustainable development of China's cultural industry as the research object and explore the influence mechanism of digital economy on the sustainable development of China's cultural industry.

[9]Bustamante, E. (2004). Cultural industries in the digital age: some provisional conclusions. Media, culture & society, 26(6), 803-820.

[10]Li, Y. (2022). Research on Performance Measurement and Promotion Countermeasures of Digital Culture Industry in Zhejiang Province. Financial Engineering and Risk Management, 5(5), 77-83.

[11]Chen, J., & Xu, S. (2022). Research on the Development of Digital Creative Sports Industry Based on Deep Learning. Computational Intelligence and Neuroscience, 2022.

[12]Shin, Y. J., & Lee, D. H. (2016). The role of the digital culture contents industry in the knowledge economy: An input-output analysis. Knowledge Management Research, 17(1), 73-89.

[13]Zhen, Z., Yousaf, Z., Radulescu, M., & Yasir, M. (2021). Nexus of digital organizational culture, capabilities, organizational readiness, and innovation: Investigation of SMEs operating in the digital economy. Sustainability, 13(2), 720.

[14]Dong, H., Bae, K. H., & Zhang, M. (2022). A Study on Technological Innovation Efficiency of Listed Companies in China’s Digital Cultural Industry. The Journal of the Korea Contents Association, 22(3), 369-379.

[15]Zhang, T., Ni, Y., & Zhang, Y. (2022). Opportunities and Challenges: An Empirical Analysis of the International Competitiveness of China. 문화산업연구, 22(2), 23-30.

[16]Duerr, S., Holotiuk, F., Wagner, H. T., Beimborn, D., & Weitzel, T. (2018). What is digital organizational culture? Insights from exploratory case studies.

[17]Sukhovskaya, D. N., & Ermakova, L. I. (2020, May). Adaptation of Culture and Creative Industries Sphere to the Transition of Digital Technologies. In 2nd International Scientific and Practical Conference “Modern Management Trends and the Digital Economy: from Regional Development to Global Economic Growth”(MTDE 2020) (pp. 474-477). Atlantis Press.

[18]Jin, J., & Zhu, J. (2018, August). Study on the Development and Evolution of Digital Content Industry and Strategic Countermeasures Taking Japan as an Example. In 3rd International Conference on Judicial, Administrative and Humanitarian Problems of State Structures and Economic Subjects (JAHP 2018) (pp. 316-320). Atlantis Press.

[19]Zhang, J., van Gorp, D., & Kievit, H. (2023). Digital technology and national entrepreneurship: An ecosystem perspective. The Journal of Technology Transfer, 48(3), 1077-1105.

  1. Include the discussion on the selection variables to be included in the model.

Response to Comment 4: 

Many thanks for pointing out to this important weakness. We supplemented the Variable Selection and Data Sources section with literature sources for the selected variables to provide theoretical support for the variables we chose. The details are as follows:

4.1. Variable Selection and Data Sources

4.1.1. Dependent Variable: Cultural Industry Development Level (CUL)

Most existing studies believe that the sustainable development of the cultural industry should include four dimensions: innovation, coordination, openness, and sharing. In order to reflect the multiple attributes of sustainable development, and consider the particularity, the quantifiability of indicators, and the availability of data. Drawing on Li et al. (2023), this paper constructs a comprehensive indicator system to measure the sustainable development level of the cultural industry from dimensions such as cultural innovation ability, coordination level, openness level, sharing level, and industrial efficiency [31]. The indicator data in the indicator system mainly comes from the National Research Network database, the Statistical Yearbook of Culture and Related Industries, and the Guotai An database; Partial missing data is calculated using trend prediction or interpolation methods. Finally, a principal component analysis was conducted on the five indicators of cultural innovation ability, collaboration level, openness, sharing level, and industrial efficiency, ultimately obtaining a comprehensive indicator.

4.1.2. Independent Variable: Digital Economy (DIG)

Drawing on Huang et al. (2019), this paper measures the level of development of the digital economy in each region from the perspectives of both Internet development and digital inclusive financial development [32]. To measure the development of the internet, this paper draws on the method of Huang Huiqun et al. (2019), choose the penetration rate of mobile phones, the employment situation in the information industry, the output situation related to the internet, and the penetration rate of the internet. To measure the development of digital financial inclusion in China, choose the digital inclusion index, and weight the five indicators collectively to determine the level of urban and regional digital inclusion.

4.1.3. Moderating Variable: Government Support (GOV)

Drawing on Wall-Andrew et al. (2021), we chose the share of expenditure on culture, sports, and media in general public budget expenditure to measure the degree of government support for development in cultural industries [33].

4.1.4. Control Variables

In order to avoid other factors affecting development in cultural industries from interfering with our empirical results, Drawing on Liu et al. (2022), we control for the following variables [34]: (1) per capita gross national product (Pgdp); (2) financial institutions’ deposit and loan balances as a percentage of the province’s gross domestic product (GDP) in that year (Finance), calculated as the ratio of the institutions’ deposit and loan balances to the GDP level of the province; (3) general public budget expenditure on culture (Budgcul); (4) culture industry endowment structure (Bingfu), as determined by the capital stock to labor force participation ratio in businesses larger than the industry; (5) the number of employees in cultural institutions (Shiye); (6) the average number of students in higher education (Educ). Finally, we controlled for province and year fixed effects. Data for each control variable were obtained from various public information sources such as the China Statistical Yearbook, China Information Industry Yearbook, and the database of the National Bureau of Statistics.

  [31]Li C. (2023). Digital Finance Helps the High Quality Development of the Cultural Industry: Mechanisms, Effects, Challenges, and Paths. Dongyue Tribune, 44, 5.

[32]Huang Q., Yu Y., & Zhang S. (2019). Internet Development and Productivity Growth in Manufacturing Industry: Internal Mechanism and China Experiences. China Ind. Econ, 8, 5–23.

[33]Wall-Andrews, C., Walker, E., & Cukier, W. (2021). Support mechanisms for Canada’s cultural and creative sectors during COVID-19. Journal of Risk and Financial Management, 14(12), 595.

[34] Liu, R., & Qiu, Z. (2022). Urban sustainable development empowered by cultural and tourism industries: Using Zhenjiang as an example. Sustainability, 14(19), 12884.

  1. Change the methodology section into methods. Include the procedure of data collection, sampling techniques and analysis.

Response to Comment 5: 

Many thanks for pointing out to this important weakness. We have changed the methodology section to methods and included the procedure of data collection, sampling techniques and analysis. See pages four through ten of the manuscript for details.

  1. Include the notation from the formula and explain how authors represent a function rule in function notation.

Response to Comment 6: 

Many thanks for pointing out to this important weakness. We have added the variables represented by the symbols in the formulas, and explained what the coefficients of the main variables represent, explaining the rules of operation of the function. The details are as follows:

4.3. Econometric Approach

This article builds two empirical models using data from 2009 to 2020. The first model examines how the digital economy affects the cultural sector’s ability to grow sustainably. This paper focuses on the coefficient  of , which reflects the impact of digital economy on the sustainable development of cultural industry. indicates the degree of impact of digital economy on the sustainable development of cultural industry.

Where: i denotes the province and t denotes the year. denotes the level of development in the cultural industry in province i in year t. denotes the level of development in the digital economy in province i in year t.  denotes the government support for development in the cultural industry in year t in province i.  denotes a set of control variables.  represents the province dummy variable.  represents the time dummy variable.  is the random disturbance term. To increase the accuracy of the regression results, we treated all of the variables as logarithms.

The second model examines how government support may mitigate the impact of the digital economy on the long-term growth of the cultural industries. This paper focuses on the interaction  between the digital economy and government support.  is the interaction term between the level of digital economy development and government support for the cultural industry, and its coefficient. denotes the extent to which the impact of the digital economy on development in the cultural industry is moderated by government support.

Where: i denotes the province and t denotes the year. denotes the level of development in the cultural industry in province i in year t. denotes the level of development in the digital economy in province i in year t.  denotes the government support for development in the cultural industry in year t in province i.  is the interaction term between the level of digital economy development and government support for the cultural industry, and its coefficient.  denotes a set of control variables.  represents the province dummy variable.  represents the time dummy variable.  is the random disturbance term. To increase the accuracy of the regression results, we treated all of the variables as logarithms.

  1. The authors need to include the classical assumptions of multiple regression.

Response to Comment 7: 

Many thanks for pointing out to this important weakness. Before performing the OLS multiple regression, we performed normality, heteroskedasticity, autocorrelation, and covariance tests to ensure that the data fit the OLS model. The details are as follows:

4.4. Empirical Analysis

4.4.1. Model Checking

We performed normality, heteroskedasticity, autocorrelation, and covariance tests before performing OLS regression to ensure that our data fit the OLS model.

Normality test: When the values of the predictor variables are fixed and the dependent variable is normally distributed, the residual values should also be a normal distribution with a mean of zero. To test for normality, we plotted the standard residual histogram and the normal P-P plot. As shown in the histogram, the standardized residuals of this regression approximated a normal distribution. Also, the points in the normal P-P plot fall essentially on the diagonal line, indicating that the regression residuals are close to a normal distribution.

Figure 1. Histogram of regression-standardized residuals

Figure 2. Normal P-P diagram of regression-standardized residuals

Heteroskedasticity test: we used Breusch-Pagan/Cook-Weisberg test for heteroskedasticity. The results show a P-value of 0.7879>0.05, so the original hypothesis (Ho: Constant variance) is not rejected, indicating that there is no heteroskedasticity problem in the data.

Autocorrelation test: We use the Lagrange multiplier test (LM test) for autocorrelation in panel data, and the result shows that the P-value is 0.0963>0.05, so the original hypothesis of LM test (Ho: No autocorrelation) is not rejected, indicating that there is no autocorrelation problem in the data. Also, it shows that the OLS model is more appropriate for the sample data than the random effects model.

Covariance test: To ensure that multicollinearity did not affect the results, we calculated variance inflation factors (VIFs). The results showed that the VIFs for all variables were below 6.03 (mean = 3.01), well below the generally accepted threshold of 10.0. We also tested the as seen in Table 2, the correlation coefficients between the variables were mostly less than 0.70, which is the lowest limit for which multicollinearity is considered possible. This indicates that the covariance problem of the data in this paper is small.

Through the above four tests, we believe that our data meet the basic conditions of OLS regression. Therefore, we choose the mixed-effects model, i.e., the POLS model.

  1. How did the authors deal with the data bias?

Response to Comment 8: 

Many thanks for pointing out to this important weakness. In order to prevent data bias from having an impact on the regression results, we treated the data as follows: first, all the variables were trimmed at the 1% level in order to avoid biasing the estimates due to the presence of extreme values. Second, before performing the OLS multiple regression, we performed normality, heteroskedasticity, autocorrelation, and covariance tests to ensure that the data conformed to the OLS model. Finally, we performed robustness tests by changing the measure of the dependent variable and rerunning the regression to ensure the robustness of the empirical results. See pages five through seven, and nine through ten of the manuscript for details.

  1. Theoretical and practical implications shall include the discussion of sustainable development and its goals.

Response to Comment 9: 

Many thanks for pointing out to this important weakness. We have added a discussion of the digital economy for the sustainable development objectives of the cultural industries in the Conclusions and Discussion section. The details are as follows:

Through the above theoretical deduction and empirical test, this paper finds that digital economy has a substantial role in promoting the sustainable development of cultural industry. The digital economy of the cultural industry has transformed the entire industry and industrial chain of the traditional cultural industry through digital technology, reflecting the multiplier and superposition impact of digital technology on promoting the development of the cultural industry. Using big data, artificial intelligence, cloud computing and other digital economy, data analysis and accurate management of the production, communication, consumption and other aspects of the cultural industry, to achieve better user experience, higher market share and better economic benefits, and further realize the digital economy to promote the sustainable development of the cultural industry. In addition, the favorable impact of the digital economy on the growth of the cultural industry has been further enhanced with the strong support of the government.

The development mode of the digital economy has become an important guarantee for maintaining economic development and an important means to promote the sustainable development of the cultural industry. To this end, the goal of promoting the sustainable development of the cultural industry by the digital economy should be clearly defined to maximize the efficiency of the digital economy. First of all, improve the management level of the cultural industry, increase the compliance construction of the cultural industry, and improve the attractiveness of the cultural industry to the digital economy. Second, strengthen the research and development of digital technology in the era of digital economy, improve the development level of digital economy, and implement the effect of digital technology to empower the cultural industry. Third, promote the integrated development of the digital economy and the cultural industry to achieve a positive interaction between the digital economy and the sustainable development of the cultural industry. Fourth, improve the supporting system for the sustainable development of the digital economy and the cultural industry, improve the regulatory system of the digital economy, and improve the governance capacity of the cultural industry. Fifth, pay attention to the cultivation of interdisciplinary talents in the field of cultural industry under the background of digital economy, and make full use of human resources to give full play to the subjective initiative of integrated development.

Once again, we really appreciate your insightful and detailed comments and instructions for improving our paper. Your great comments inspire us a lot to streamline our overall paper and make it looks much coherent and interesting than the last version. We sincerely hope that our revision has adequately addressed all the issues you raised. Thank you!

Best wishes,

Qingjin Wang

Reviewer 2 Report (Previous Reviewer 3)

Thank you for the update and modification.

Author Response

Response Letter to the Comments of Reviewer #2

“How the Digital Economy Enables Regional Sustainable Development using Big Data Analytics”

Submitted to

Sustainability

Dear Reviewer #2,

Thank you very much for your approval of our article. We have made the following changes and improvements from the previous version to make the article richer, more logically rigorous (We have highlighted the changes in red in the article).

First, we have added a literature review section, in which we compare the sustainable development of China's cultural industry with other countries. It also clarifies why it is important to conduct the study in the Chinese context.

Second, before performing the OLS multiple regression, we performed normality, heteroskedasticity, autocorrelation, and covariance tests to ensure that the data fit the OLS model.

Third, we performed robustness tests by changing the measure of the dependent variable and rerunning the regression to ensure the robustness of the empirical results.

Fourth, we have added a discussion of the digital economy for the sustainable development objectives of the cultural industries in the Conclusions and Discussion section.

Once again, thank you for your contribution to the revision of our manuscript.

Best wishes,

  Qingjin Wang

Round 2

Reviewer 1 Report (Previous Reviewer 2)

This paper can be accepted in its present form. 

This manuscript is a resubmission of an earlier submission. The following is a list of the peer review reports and author responses from that submission.

Round 1

Reviewer 1 Report

Although the structure of the article serves the purpose of this article, the main problem of the article is related to the lack of heterogeneity in the data. We have no information on the heterogeneity of cultural institutions from the perspective of ownership and control (public or private). If the government controls both sides of the process (building digital infrastructure and supporting cultural industries), the correlation must be positive, especially in a country like China. Moreover, if cultural industries are controlled by the government, they serve the government, so they use digitization as an instrument. 

A positive correlation between digitization defined by various infrastructure numbers must serve the process of developing services (cultural industries) based on this infrastructure, especially when both processes are controlled by the government (financial support). Moreover, relying on quantitative factors (the number of cultural institutions) we are unable to distinguish between high and medium quality cultural industry development. Therefore, hypothesis 1 is too trivial to consider. In short, if the government supports both processes (development of digital infrastructure and cultural institutions), the correlation must be positive. 

On the other hand, if we assume that the government is the main or even the only player supporting the development of digitization and the development of cultural industries, the correlation between the two must again be positive. Thus, all the suggestions, including Building a Talent Training System for the Digital Culture Industry, Strategy for the Future Development of the Digital Culture Industry and Promoting the Development of High-Quality Digital Cultural Products, become nothing more than a propaganda manifesto in which the government, under the guise of supporting digitization, controls all aspects of the cultural industry, using digitization as a tool.

Therefore, I propose to reject this article.

Author Response

Response Letter to the Comments of Reviewer #1

“How the Digital Economy Enables Regional High-Quality Development using Big Data Analytics”

Submitted to

Sustainability

Dear Reviewer #1,

We would like to thank you for the comments to this paper. We hope that you find it considerably improved. In the revision, we have strived as much as possible to address all the issues raised by you. Below, we provide detailed responses to your comments (We have highlighted the changes in red in the article).

  1. Although the structure of the article serves the purpose of this article, the main problem of the article is related to the lack of heterogeneity in the data. We have no information on the heterogeneity of cultural institutions from the perspective of ownership and control (public or private). If the government controls both sides of the process (building digital infrastructure and supporting cultural industries), the correlation must be positive, especially in a country like China. Moreover, if cultural industries are controlled by the government, they serve the government, so they use digitization as an instrument.

Response to Comment 1: 

Many thanks for pointing out to this important weakness. We reconstructed a comprehensive index system to measure the level of high-quality development of cultural industries in terms of cultural innovation capacity, coordination level, openness, sharing level and industrial efficiency (see Table 1 for details). This comprehensive index system of the high quality development level of cultural industry includes several subjects such as cultural enterprises, cultural manufacturing industry, per capita cultural consumption expenditure, public cultural infrastructure and cultural and art research institutions, rather than only consisting of public cultural infrastructure established by the government, which can better measure the high quality development level of cultural industry. At the same time, this comprehensive index system of cultural industry quality development level also indicates that the cultural industry is composed of multiple subjects, following the law of market development, and not only controlled by the government. This helps to more objectively test the relationship between the digital economy and the high quality development of cultural industries.

Table 1 Indicator system for measuring the level of high-quality development of cultural industries

Primary indicators

Secondary indicators

Properties

Innovation Capability

Number of culture, art, science and technology, scientific research institutions

+

Culture, art, science and technology, research institutions assets

+

Number of professional and technical personnel in cultural research institutions

+

R&D investment intensity

+

Number of patents obtained by cultural enterprises

+

Number of works copyrighted by cultural enterprises

+

Number of software copyrights obtained by cultural enterprises

+

Coordination level

Ratio of per capita cultural and entertainment consumption expenditure of urban and rural residents

+

Advanced industrial structure

+

Degree of openness

Number of participants in foreign cultural exchange activities

+

Number of foreign cultural exchange projects

+

Sharing Level

Public library holdings per capita

+

Public library floor space per capita

+

Museum collections per 10,000 people

+

Industry Benefits

Operating profit of cultural manufacturing, wholesale and retail, and service industries above the scale

+

Above-scale cultural manufacturing, wholesale and retail, and services business taxes and surcharges

+

Number of cultural activities organized by mass cultural institutions

+

Number of participants in mass cultural institutions and libraries

+

Museum Visits

+

Number of people attending lectures at mass cultural institutions and libraries

+

  1. A positive correlation between digitization defined by various infrastructure numbers must serve the process of developing services (cultural industries) based on this infrastructure, especially when both processes are controlled by the government (financial support). Moreover, relying on quantitative factors (the number of cultural institutions) we are unable to distinguish between high and medium quality cultural industry development. Therefore, hypothesis 1 is too trivial to consider. In short, if the government supports both processes (development of digital infrastructure and cultural institutions), the correlation must be positive.

Response to Comment 2: 

Many thanks for pointing out to this important weakness. We reconstructed a comprehensive index system to measure the level of high-quality development of cultural industries in terms of cultural innovation capacity, coordination level, openness, sharing level and industrial efficiency (see Table 1 for details). This comprehensive index system of the high quality development level of cultural industry includes several subjects such as cultural enterprises, cultural manufacturing industry, per capita cultural consumption expenditure, public cultural infrastructure and cultural and art research institutions, rather than only consisting of public cultural infrastructure established by the government, which can better measure the high quality development level of cultural industry. At the same time, this comprehensive index system of the high quality development level of cultural industry includes not only the number of cultural institutions, but also the innovation ability of cultural enterprises, per capita cultural consumption, and the operating income of cultural manufacturing industry, which is a more comprehensive measure of the high quality development level of cultural industry from five perspectives: cultural innovation ability, coordination level, openness, sharing level, and industrial efficiency. In addition, we have modified hypothesis one to more accurately represent the relationship between the digital economy and the high-quality development of cultural industry, making its logic clearer.

  1. On the other hand, if we assume that the government is the main or even the only player supporting the development of digitization and the development of cultural industries, the correlation between the two must again be positive. Thus, all the suggestions, including Building a Talent Training System for the Digital Culture Industry, Strategy for the Future Development of the Digital Culture Industry and Promoting the Development of High-Quality Digital Cultural Products, become nothing more than a propaganda manifesto in which the government, under the guise of supporting digitization, controls all aspects of the cultural industry, using digitization as a tool.

Response to Comment 3: 

Many thanks for pointing out to this important weakness. We reconstructed a comprehensive index system to measure the level of high-quality development of cultural industries in terms of cultural innovation capacity, coordination level, openness, sharing level and industrial efficiency (see Table 1 for details). This comprehensive index system of the high quality development level of cultural industry includes several subjects such as cultural enterprises, cultural manufacturing industry, per capita cultural consumption expenditure, public cultural infrastructure and cultural and art research institutions, rather than only consisting of public cultural infrastructure established by the government, which can better measure the high quality development level of cultural industry. At the same time, this comprehensive index system of cultural industry quality development level also indicates that the cultural industry is composed of multiple subjects, following the law of market development, and not only controlled by the government. This helps to more objectively test the relationship between the digital economy and the high quality development of cultural industries.

In addition, we used the reconstructed comprehensive index system of the level of high-quality development of cultural industries to re-test the empirical part.  The empirical results show that the digital economy has a positive and significant impact on the high-quality development of the cultural industry; government support has a moderating effect in the process of the digital economy influencing the high-quality development of the cultural industry. The empirical results remain consistent with our proposed view, and H1 and H2 are verified.

Once again, we really appreciate your insightful and detailed comments and instructions for improving our paper. Your great comments inspire us a lot to streamline our overall paper and make it looks much coherent and interesting than the last version. We sincerely hope that our revision has adequately addressed all the issues you raised. Thank you!

Best wishes,

Qingjin Wang

Reviewer 2 Report

The manuscript can comply with the previous comments and meet the academic standard. The issue has been linked to the methodological section, results and implications. The minor improvement, if possible, to justify the novelty of this study compared to existing ones. Good luck to the authors. 

Author Response

Response Letter to the Comments of Reviewer #2

“How the Digital Economy Enables Regional High-Quality Development using Big Data Analytics”

Submitted to

Sustainability

Dear Reviewer #2,

Thank you very much for your approval of our article. We have made the following changes and improvements from the previous version to make the article richer, more logically rigorous (We have highlighted the changes in red in the article).

First, we reconstructed a comprehensive index system to measure the level of high-quality development of cultural industries in terms of cultural innovation capacity, coordination level, openness, sharing level and industrial efficiency (see Table 1 for details). This comprehensive index system of the high quality development level of cultural industry includes several subjects such as cultural enterprises, cultural manufacturing industry, per capita cultural consumption expenditure, public cultural infrastructure and cultural and art research institutions, rather than only consisting of public cultural infrastructure established by the government, which can better measure the high quality development level of cultural industry. At the same time, this comprehensive index system of the high quality development level of cultural industry includes not only the number of cultural institutions, but also the innovation ability of cultural enterprises, per capita cultural consumption, and the operating income of cultural manufacturing industry, which is a more comprehensive measure of the high quality development level of cultural industry from five perspectives: cultural innovation ability, coordination level, openness, sharing level, and industrial efficiency. In addition, we have modified hypothesis one to more accurately represent the relationship between the digital economy and the high-quality development of cultural industry, making its logic clearer.

Table 1 Indicator system for measuring the level of high-quality development of cultural industries

Primary indicators

Secondary indicators

Properties

Innovation Capability

Number of culture, art, science and technology, scientific research institutions

+

Culture, art, science and technology, research institutions assets

+

Number of professional and technical personnel in cultural research institutions

+

R&D investment intensity

+

Number of patents obtained by cultural enterprises

+

Number of works copyrighted by cultural enterprises

+

Number of software copyrights obtained by cultural enterprises

+

Coordination level

Ratio of per capita cultural and entertainment consumption expenditure of urban and rural residents

+

Advanced industrial structure

+

Degree of openness

Number of participants in foreign cultural exchange activities

+

Number of foreign cultural exchange projects

+

Sharing Level

Public library holdings per capita

+

Public library floor space per capita

+

Museum collections per 10,000 people

+

Industry Benefits

Operating profit of cultural manufacturing, wholesale and retail, and service industries above the scale

+

Above-scale cultural manufacturing, wholesale and retail, and services business taxes and surcharges

+

Number of cultural activities organized by mass cultural institutions

+

Number of participants in mass cultural institutions and libraries

+

Museum Visits

+

Number of people attending lectures at mass cultural institutions and libraries

+

Second, we used the reconstructed comprehensive index system of the level of high-quality development of cultural industries to re-test the empirical part.  The empirical results show that the digital economy has a positive and significant impact on the high-quality development of the cultural industry; government support has a moderating effect in the process of the digital economy influencing the high-quality development of the cultural industry. The empirical results remain consistent with our proposed view, and H1 and H2 are verified.

Third, we have added sections to the introduction, research hypothesis, and conclusion and discussion sections to describe in more detail the relationship between the digital economy and the high-quality development of cultural industries.

Fourth, we have carefully checked and sorted out the references and replaced some of them to make them correspond to the content of the article.

Finally, we have used the editing service (https://www.mdpi.com/authors/english) to proofread our articles to make the language more accurate and smooth.

Once again, thank you for your contribution to the revision of our manuscript.

Best wishes,

  Qingjin Wang

Reviewer 3 Report

thank you for the improvements.

Author Response

对审稿人评论的回复信 #3

“数字经济如何利用大数据分析使能区域高质量发展”

提交给

可持续性

亲爱的审稿人 #3,

非常感谢您对我们文章的认可。我们对以前的版本进行了以下更改和改进,以使文章更丰富,逻辑更严谨(我们在文章中以红色突出显示了更改)。

首先,构建综合指标体系,从文化创新能力、协调水平、开放程度、共享水平、产业效益等方面衡量文化产业高质量发展水平(详见表1)。这个文化产业高质量发展水平的综合指标体系包括文化企业、文化制造业、人均文化消费支出、公共文化基础设施和文化艺术研究机构等几个学科,而不仅仅是由政府建立的公共文化基础设施组成,更能衡量文化产业高质量发展水平。同时,这一文化产业高质量发展水平的综合指标体系,不仅包括文化机构数量,还包括文化企业的创新能力、人均文化消费、文化制造业营业收入,从五个角度更全面地衡量文化产业高质量发展水平: 文化创新能力、协调水平、开放性、共享性、产业效益。此外,我们还修改了假设一,以更准确地表示数字经济与文化产业高质量发展的关系,使其逻辑更加清晰。

表1衡量文化产业高质量发展水平的指标体系

主要指标

次要指标

性能

创新能力

文化、艺术、科技、科研机构数量

+

文化、艺术、科技、科研机构资产

+

文化研究机构专业技术人员人数

+

研发投入强度

+

文化企业获得的专利数量

+

文化企业著作权作品数量

+

文化企业获得的软件著作权数量

+

协调级别

城乡居民人均文化娱乐消费支出比值

+

先进产业结构

+

开放程度

参加对外文化交流活动的人数

+

对外文化交流项目数量

+

共享级别

人均公共图书馆馆藏

+

人均公共图书馆建筑面积

+

每万人的博物馆藏品

+

Industry Benefits

文化制造、批发零售、规模以上服务业营业利润

+

Above-scale cultural manufacturing, wholesale and retail, and services business taxes and surcharges

+

Number of cultural activities organized by mass cultural institutions

+

Number of participants in mass cultural institutions and libraries

+

Museum Visits

+

Number of people attending lectures at mass cultural institutions and libraries

+

Second, we used the reconstructed comprehensive index system of the level of high-quality development of cultural industries to re-test the empirical part.  The empirical results show that the digital economy has a positive and significant impact on the high-quality development of the cultural industry; government support has a moderating effect in the process of the digital economy influencing the high-quality development of the cultural industry. The empirical results remain consistent with our proposed view, and H1 and H2 are verified.

Third, we have added sections to the introduction, research hypothesis, and conclusion and discussion sections to describe in more detail the relationship between the digital economy and the high-quality development of cultural industries.

Fourth, we have carefully checked and sorted out the references and replaced some of them to make them correspond to the content of the article.

Finally, we have used the editing service (https://www.mdpi.com/authors/english) to proofread our articles to make the language more accurate and smooth.

Once again, thank you for your contribution to the revision of our manuscript.

Best wishes,

  Qingjin Wang

Round 2

Reviewer 1 Report

Thank you for resubmitting your article. Your new comments help to better understand your perspective. I can now recommend your article for publication.